# Reduction in Egg Fertility of *Aedes albopictus* Mosquitoes in Greece Following Releases of Imported Sterile Males

**DOI:** 10.3390/insects12020110

**Published:** 2021-01-27

**Authors:** Georgios Balatsos, Arianna Puggioli, Vasileios Karras, Ioanna Lytra, George Mastronikolos, Marco Carrieri, Dimitrios P. Papachristos, Marco Malfacini, Angeliki Stefopoulou, Charalampos S. Ioannou, Fabrizio Balestrino, Jérémy Bouyer, Dušan Petrić, Igor Pajović, Apostolos Kapranas, Nikos T. Papadopoulos, Panagiotis G. Milonas, Romeo Bellini, Antonios Michaelakis

**Affiliations:** 1Scient. Directorate of Entomology and Agricultural Zoology, Benaki Phytopathological Institute, 14561 Kifissia, Greece; g.balatsos@bpi.gr (G.B.); v.karras@bpi.gr (V.K.); i.lytra@bpi.gr (I.L.); d.papachristos@bpi.gr (D.P.P.); a.stefopoulou@bpi.gr (A.S.); a.kapranas@bpi.gr (A.K.); p.milonas@bpi.gr (P.G.M.); 2Centro Agricoltura Ambiente “G. Nicoli”, 40014 Crevalcore, Italy; apuggioli@caa.it (A.P.); mcarrieri@caa.it (M.C.); mmalfacini@caa.it (M.M.); fbalestrino@caa.it (F.B.); rbellini@caa.it (R.B.); 3Department of Agriculture, Crop Production and Rural Environment, University of Thessaly, 38446 Magnisias, Greece; mastronikgeo@ymail.com (G.M.); ioannoubabis@yahoo.com (C.S.I.); nikopap@uth.gr (N.T.P.); 4Insect Pest Control Laboratory, Joint FAO/IAEA Programme of Nuclear Techniques in Food and Agriculture, Seibersdorf, A-2444 Vienna, Austria; J.Bouyer@iaea.org; 5Faculty of Agriculture, University of Novi Sad, 21000 Novi Sad, Serbia; dusan.petric@polj.uns.ac.rs; 6Biotechnical Faculty, University of Montenegro, 81000 Podgorica, Montenegro; pajovicb.igor@gmail.com

**Keywords:** mosquito control, sterile insect technique, egg hatch rate, Asian tiger mosquito, mass production, transportation

## Abstract

**Simple Summary:**

*Aedes albopictus*, also known as the Asian tiger mosquito, is an invasive mosquito well established in Europe, posing high risks of transmission of dengue and chikungunya viruses that are detrimental to human health. The newly invaded areas that experienced no intense mosquito annoyance in the past are now suffering because of the Asian tiger mosquito. Due to the large number of diverse breeding sites and the poor community participation in mosquito habitat elimination programs, traditional control efforts, such as intense chemical control efforts in public areas, have rather low efficacy. The sterile insect technique (SIT) is a method of insect control with successful applications against several agricultural insect pests and it has been proposed as a control method against invasive mosquito species. Weekly release of 2280 to 2995 sterile male mosquitoes/hectare resulted in increased levels of egg sterility of *Ae. albopictus* at a single release site of 5 hectares. This first SIT pilot trial against *Ae. albopictus* in Greece showed encouraging results that justify the continuation with larger scale trials.

**Abstract:**

*Aedes albopictus*, an invasive mosquito species, is currently well established in many European countries, posing high risks to human health. A preliminary trial using repetitive releases of irradiated sterile male mosquitoes was designed, implemented and evaluated for the first time in Greece. The main scope of this trial was to investigate the efficacy of sterile insect technique (SIT) on wild population egg hatch rate in Greece using mass-produced sterile male mosquitoes from another country (Italy). The study was conducted in Vravrona area, close to Athens International Airport (Attica Region). The number of eggs laid in ovitraps was similar in all intervention and control plots. However, a significant reduction in egg hatch rate was recorded in the SIT plot in comparison with both control plots starting two weeks after the first release. This trial validates the logistics (transportation, releases handling and monitoring) as a major step towards implementing efficient, environmentally safe control approaches as an additional tool against the invasive *Aedes* species in Greece and more widely in Europe.

## 1. Introduction

The introduction of *Aedes* invasive mosquito species to new areas increases the nuisance caused by mosquitoes and more importantly the transmission risk of mosquito-borne diseases [1]. Risks for human health involving *Aedes* mosquito vectors include chikungunya, dengue and Zika viruses especially in urban areas where the vector as well as human densities are higher [2]. In Greece, the Asian tiger mosquito, *Aedes (Stegomyia) albopictus* (Skuse), was reported for the first time in 2003 in Thesprotia and Corfu (northwest Greece) [3]. Since then, *Ae. albopictus* has invaded most of the continental part of the country except some continental areas of northern Greece and a few islands of the Aegean Sea [4]. *Aedes albopictus* is a container-breeding mosquito species mainly developing in urban areas with vegetation. Traditional control efforts using insecticides in public areas have rather low efficacy mainly because of the large number of diverse breeding sites and poor community participation in mosquito habitat elimination programs. Urban areas include a large number of private properties providing a variety of breeding sites which are often not accessible by pest control operators [5].

In recent years, the Asian tiger mosquito caused intense nuisance in areas that in the past did not experience noteworthy mosquito problems. Regardless of the control efforts, it is presently almost impossible to eliminate the local risk of transmission of exotic arboviruses once an *Aedes* species is established in an area [2,6,7,8]. The Greek dengue epidemic of 1927–1928 was one of the largest dengue fever epidemics in Europe transmitted by *Ae. aegypti* mosquitoes which is not present in Greece anymore. Nowadays, the only invasive mosquito species (IMS) that is present in Greece is *Ae. albopictus* and its establishment may cause problems for public health [9,10,11]. Therefore, a management plan to control *Ae*. *albopictus* has been structured as a comprehensive practical technical guideline to assist local authorities [12]. The current management plan includes coordinated actions such as monitoring protocols and activities for stakeholders and local communities. The implementation of new control approaches, such as the sterile insect technique (SIT), Wolbachia-based strategies, pyriproxyfen autodissemination and transgenic technologies have recently been proposed [6,13]. SIT is a method of insect control with a strong record of successful applications against a range of agricultural insect pests [14,15]. The limitations of current vector control strategies especially in areas with mosquito-borne diseases [16,17] and the availability of new equipment that supports the implementation of SIT (e.g., mass-rearing cages for adults and racks for larvae, sex sorters, automatic release systems by drones) [18,19] have sparked interest in utilizing this approach. Pilot SIT field trials conducted against *Ae. albopictus* in Italy demonstrated the feasibility of this approach and paved the way for scaling up and expanding its implementation [16,17]. A successful recent field trial in China demonstrated the feasibility of area-wide application of a combination of an incompatible insect technique (caused by the maternally inherited endosymbiotic bacteria Wolbachia) and SIT for *Ae. albopictus* control [20,21].

In this study, we designed, implemented and evaluated SIT technique in a pilot trial in Greece as a tool to control the Asian tiger mosquito. Our main goal was to evaluate the efficacy of the method in Greece by importing mass-produced mosquitoes from a facility in another country (Italy). This pilot study represents a preliminary phase for a larger scale trial and therefore we tried to test and validate the logistics and release handling using mass-produced sterile male mosquitoes from another country (Italy). Moreover, we evaluated the effect of sterile male mosquito releases on egg fertility in the targeted trial area. 

## 2. Materials and Methods 

### 2.1. The Trial Area

The trial took place in Vravrona, Markopoulo municipality, Attica region of Greece, located east of Athens. The area was selected based on (a) its accessible size and its ecological isolation and (b) its close proximity (15 km) to the Athens International Airport, where sterile males were delivered. Three plots, 5 ha each, were defined: SIT, control A and control B. 

The pilot trial was conducted in an urban locality of 5 ha surface, determined by our supply capacity of 3000 sterile males/ha. The total area of the Vravrona site was almost 10 ha and it was isolated from other urban areas since the surrounding area comprised high hills covered by montane forests. The area is surrounded by sea on the north side, while on the south, west and east side, the nearest urban areas are located at a distance of almost 1.5 km. This area was divided into two equal parts of 5 hectares each; one plot served as intervention area (SIT, 37°55′13.12″ N, 24°00′42.51″ E) and the other as a distant control area (control A, 37°55′05.18″ N, 24°00′39.17″ E) (Figure 1). A third plot (control B, 37°55′7.20″ N, 24°1′19.81″ E) was also included in our trial (Figure 1). The control B plot was 1.5 km away from the intervention plots and all plots were characterized by the same pattern of small family houses and private gardens with Mediterranean vegetation. 

In each plot, a temperature–humidity logger with internal sensor (type Ebro EBI 20 TH1, Xylem Analytics Germany Sales GmbH & Co) was activated. Records of temperature and relative humidity (RH) are given in Appendix A. Additional data on the rainfall, wind speed and daylength for the period from 10 September 2018 to 25 November 2018 (weeks No. 37 to No. 47) are available in Appendix A.

### 2.2. Information Campaign and Public Education 

Before releasing the sterile males, an education campaign took place aiming to inform the residents about this technique and also to raise community participation in mosquito habitat elimination programs. Moreover, due to the fact that our capacity was limited to 3000 sterile males/ha, we decided to implement a public education campaign aiming to reduce breeding sites in private areas (door-to-door campaign—DtoD).

The door-to-door campaign was implemented in the whole Vravrona area (SIT and control A plots) prior to the SIT trial and concluded one month before implementing the first sterile male release (from 20 July until 14 August 2018). Residents were informed about the concept and benefits of SIT and about the ongoing sterile mosquito releases. In parallel, we provided advice on how to eliminate mosquito breeding sites on their properties. The visual support (leaflet) used for these educational activities was the same as we used in a previous study in Greece that briefly explained the Asian tiger mosquito life cycle and potential breeding sites [5]. All households were visited twice: in the first visit, the residents were informed about the scope of the trials and in the second visit, they were instructed how to identify and eliminate the breeding sites themselves. Informing local residents was an important component of the project since this was the first time that SIT had been implemented against mosquitoes in Greece. 

### 2.3. Mosquito Mass Rearing and Sterilization 

The mosquito colony used for the SIT pilot study originated from eggs collected in Vravrona in 2018 (June and July), transferred and mass-reared at the Centro Agricoltura Ambiente (CAA) “G. Nicoli”, Italy, following methods previously described in Bellini et al. [17] and Balestrino et al. [22,23]. Irradiation was performed by exposing 24–36-h old pupae, kept in water, to a Cesium-137 source at a dose rate of 2.2 Gy/min ± 3.5% [22]. The gamma ray dose used was 35 Gy, which resulted in a sub-sterilization in exposed males, with a residual fertility of 1.36% (±0.45 SD). Irradiated pupae were left to emerge in a climatic chamber and packaging of chilled adult sterile males followed International Air Transport Association (IATA) regulations. 

### 2.4. Sterile Mosquito Transport and Releases

Sterile males were released every Friday, from 14 September to 6 November 2018 except for 24 September when the shipment arrival was unexpectedly delayed for three days, causing high mortality. Transportation time from CAA “G. Nicoli”, Italy, to the study area was tested through a preliminary shipment and was determined to be less than 18 h. Table 1 summarizes the number of mosquitoes delivered every week, the mortality rate and the weekly estimated density of sterile males released per hectare. Each mosquito box contained 1000–1500 sterile males. During transportation, a 5% sugar in water solution was supplied in each box with a filter paper wick (Whatman chromatography paper). The sterile males were released approximately two hours after receipt by opening the release boxes in the field area. Each mosquito box was taken to the release point by foot following predefined paths in order to cover the whole plot. The predefined paths followed the positions of ovitraps (average distance 50 m).

### 2.5. Assessment of Egg Hatch

In each of the three 5 ha plots, 15 ovitraps were placed and sampled at weekly intervals starting three weeks before the first release and stopping when zero eggs were collected in all traps for two consecutive weeks (end of November). The collected oviposition substrates were kept wet using wet filter paper to wrap them during transport. In the laboratory, the oviposition substrates were left for 1 day to dry (laboratory conditions: 25 ± 1 °C, 80% RH, 14:10 L:D). The next day they were placed in a container with potassium sulfate (K_2_SO_4_) saturated solution to maintain proper humidity which keeps the eggs alive for several months [24]. Eggs were stored for embryonation for a minimum of 4 days. The protocol of Balestrino et al. 2010 [22] was adopted and egg hatch was determined by counting hatched and unhatched eggs. Since there is only one IMS established in the area, all of the obtained adults were identified as *Ae. albopictus* (*Ae. aegypti*, *Ae. japonicus* and *Ae. koreicus* are not present in Greece).

### 2.6. Statistical Analysis

To analyze the impact of the release of sterile males on the hatch rate, binomial linear mixed effect models were used, with the trapping sites as random effects and the week and treatment (SIT versus control A or B) as fixed effects [25,26]. 

To analyze the impact of the release of sterile males on egg density (number of eggs per trap per week), Poisson linear mixed effect models were used, with the trapping sites as random effects and the week and treatment (SIT versus control A or B) as fixed effects [25,26]. This was done before the start of the releases to compare the three study sites as well as after the start of the releases to compare the treatments. The quality of the models was assessed using the modified Akaike information criterion [27].

## 3. Results

Temperature and relative humidity were almost identical in all three plots throughout the field trials period (Appendix A). Relative humidity ranged from 60 to 80%, while daily average temperature ranged from 15 to 25 ℃, showing a progressive decrease from September to November. The study period (37–47 weeks) was characterized by a period of drought, interrupted by intense rains only at the end of September (week 39) and after mid-November (week 46). Before sterile mosquito releases (weeks 34, 35 and 36), the control plot A had a lower egg density as compared to the SIT (*p* = 0.0132, Appendix A) and a marginally lower one as compared to the SIT plot B (*p* = 0.159), but hatching rates were similar in all three plots (*p* > 0.857, Appendix A, Table 2). 

Between 2280 and 2995 sterile males were released per hectare per week. Mortality recorded during transportation varied in the range 5–24.9% with a median value of 15.1% (Table 1). 

Overall, there were significant differences in the average number of eggs collected in the ovitraps among different plots (Figure 2). The density of eggs in the release site remained similar to that of control A (*p* = 0.257, Appendix A) and control B (*p* = 0.09). After the beginning of the releases in the SIT site, the median egg hatch rate dropped significantly (*p* < 10^−3^, Figure 3, Appendix A), fluctuating from 14 (week 43) to 54% (week 44), showing consistent reduction related to the control plots but remaining similar in the two control sites (*p* = 0.964). 

## 4. Discussion

*Aedes albopictus* is well established in Europe, and outbreaks of diseases caused by dengue and chikungunya viruses have been reported in Europe since 2007 [28]. Southern Greece is considered suitable for the establishment of *Ae. aegypti* and other important tropical invasive mosquitoes [29,30]. The scientific community working on invasive mosquito species agrees that new control approaches and strategies are needed [31,32]. Therefore, it is of high importance to further evaluate SIT as an extra tool against *Aedes* invasive mosquito species in residential areas—areas with a high risk of vector introduction, establishment, and disease transmission [31,32]. Due to its safety and its applicability, even in the case of insecticide resistance in mosquitoes, use of SIT as a tool in integrated vector control has gained increasing international interest. This method is being evaluated against different species of mosquitoes in 34 different regions worldwide [18]. In addition, it will be tested soon as part of global health efforts to control diseases such as chikungunya, dengue, and Zika in tropical countries, with the aim of reducing vector densities below the threshold for both nuisance and disease transmission [31]. 

One main limitation in the area-wide application of SIT against mosquitoes is the high production cost of sterile males required to achieve reduction in the wild population densities. Larval control and reduction in breeding sites, both before and during the release of sterile males, is recommended to make SIT sustainable in terms of cost-efficacy. Our approach demonstrated that weekly release of 2280 to 2995 sterile male mosquitoes/hectare may induce high levels of egg sterility in a local population of *Ae. albopictus*. The induced sterility was observed as of the second week following the first release and remained at high levels throughout the rest of the season in SIT area, fluctuating in the range of 40–84% without showing a clear decreasing trend. This is in agreement with previous SIT studies conducted in Italy, which demonstrated that when the induced egg sterility rate of the Asian tiger mosquito was less than 40–50%, there was no clear reduction in the egg density [33]. In the SIT plot, mean egg hatch rates fluctuated in the range of 14–54% during the eight weeks of the trial, showing consistent reduction related to the control plots. However, using the number of eggs laid in oviposition traps as a proxy to estimate the density of the feral mosquito population, we did not observe any suppression in response to SIT application despite the high levels of induced sterility. The lack of significant impact on egg density that was observed could probably be linked to the short period of the trial and to the strong increase in the population following the rain that, probably, led to a reduction in the sterile/wild male ratio, reinforcing the effectiveness of the SIT [34,35]. 

One of the main limitations was the production capacity of males to be released per ha (available max 3000). Therefore, it was not possible to replicate the study. However, the results are clear and meaningful regarding induced sterility, although baseline data for population abundance and dynamics were not available and therefore, we cannot consider that the sites had similar wild population densities. Another limitation was the small size of the pilot area which may have suffered from the immigration of already mated females, a phenomenon which is difficult to evaluate. Finally, even if shipment time was relatively short (less than one day), moderate mortality rates were observed (Table 1). Further studies will seek to experimentally confirm any sterile male quality issues emanating from production and transportation/release of sterile mosquitoes. 

Although it is largely assumed that *Ae. albopictus* females in urbanized areas had a low dispersal capacity [33,36,37], recent mark–release–recapture studies conducted in urbanized areas in southern Switzerland revealed that this species can disperse more than 250 m [38]. One of the constraints of our trial was the small size of the plots which possibly favored the immigration of already mated females from the surrounding areas into the SIT area, a parameter which is difficult to evaluate [16,33]. 

## 5. Conclusions

This first SIT pilot trial against *Ae. albopictus* in Greece showed encouraging results that justified the continuation with larger pilot trials. We envisaged many limitations which may have affected our trial, especially for reducing egg density in the SIT plot. Besides the lack of baseline survey, our study demonstrates that SIT could be used as part of an area-wide integrated pest management program following a phase-conditional approach, that is, from preparatory activities to operational deployment, with some highlighted milestones that include go/no-go criteria [18]. A major limitation for scaling up the size of the field trial is the stock of sterile males required, which are not readily available in Greece. Our promising results may support an investment for a local mass production facility which will help us to overcome the abovementioned limitations.

## Figures and Tables

**Figure 1 insects-12-00110-f001:**
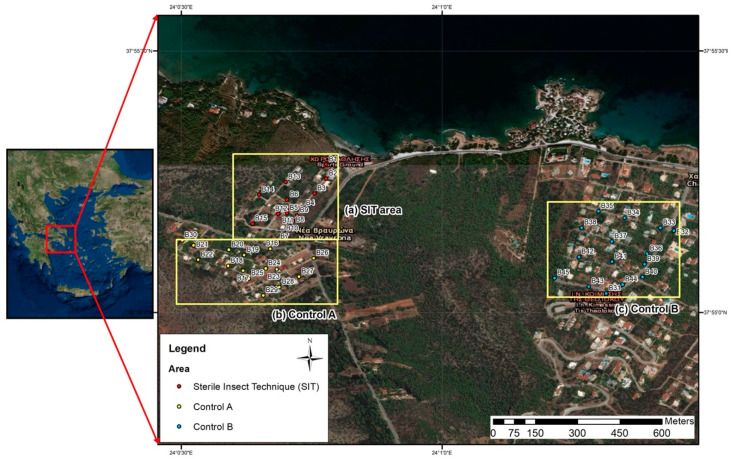
Magnified map showing the three plots: sterile insect technique (SIT), control A and control B. Dots (red, yellow and blue) on the plots indicate the positions of ovitraps. The standard operational procedures for ovitrap field management are available in Bellini et al., 2020 [12].

**Figure 2 insects-12-00110-f002:**
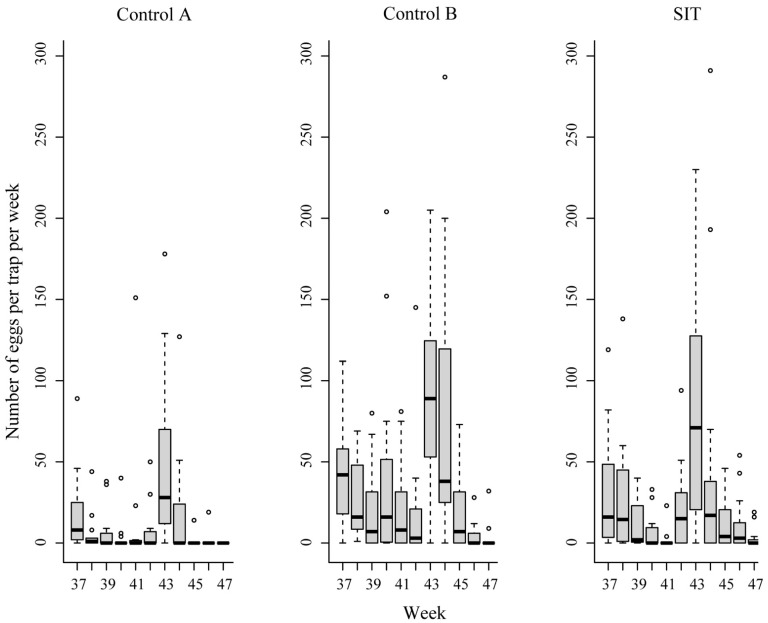
Weekly average number of eggs per ovitrap in control A, control B and the SIT area. Each plot had 15 ovitraps (3 traps/ha). Box and whisker plots present the median value (central bold line), the quartiles (box), the 95% percentiles (horizontal bars) and the outliers (dots).

**Figure 3 insects-12-00110-f003:**
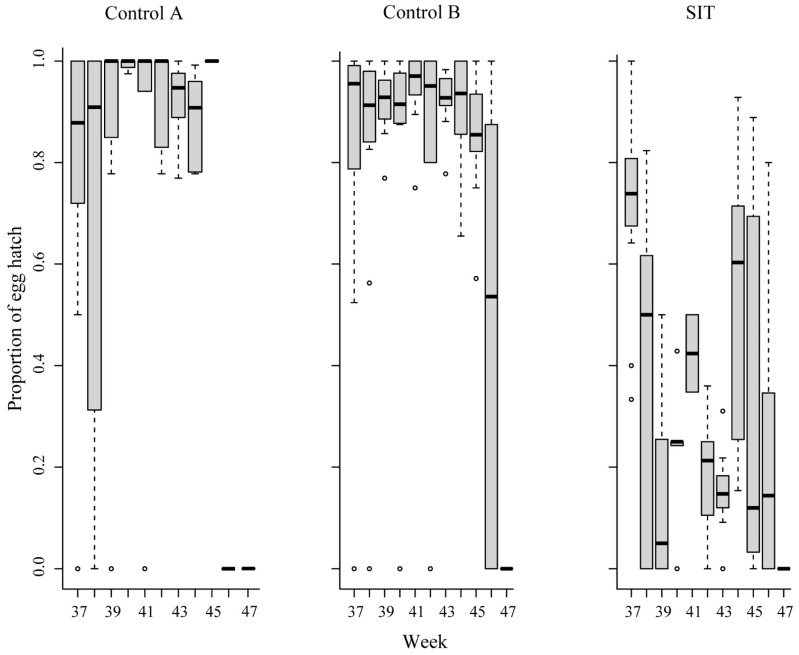
Weekly proportion of egg hatching in control A, control B and SIT plots. Box and whisker plots present the median value (central bold line), the quartiles (box), the 95% percentiles (horizontal bars) and the outliers (dots).

**Table 1 insects-12-00110-t001:** Number of males delivered, mortality rate during transport and number of sterile males released in the SIT plot.

Date	No of Males Delivered	Mortality during Transport (%).	Estimated Number of Sterile Males Released/Ha/Week
14 September 2018	15,000	12.7	2619
28 September 2018	18,000	16.8	2995
5 October 2018	15,000	15.1	2547
12 October 2018	19,000	24.9	2853
19 October 2018	12,000	5.0	2280
26 October 2018	15,000	10.4	2688
6 November 2018	17,000	24.7	2560

**Table 2 insects-12-00110-t002:** Hatching rate (%), total number of eggs/plot/week and average number of eggs/trap (±SE) examined in each plot for weeks 34, 35 and 36 (period before sterile mosquito releases). Each plot had 15 ovitraps (3 traps/ha).

Plots	Results Related To Eggs	No of Week (Period)		
		34 (24–31 August 2018)	35 (31 August–7 September 2018)	36 (7–14 September 2018)
SIT	Hatching rate (%)	92.5	96.5	89.9
	Total number eggs/plot/week	1755	1388	951
	Average No. of eggs/trap (±SE)	58.5 (±22.1)	46.3 (±20.8)	31.7 (±10.9)
Control A	Hatching rate (%)	92.1	94.4	80.0
	Total number eggs/plot/week	784	649	310
	Average No. of eggs/trap (±SE)	52.3 (±12.8)	43.3 (±10.1)	20.7 (±6.5)
Control B	Hatching rate (%)	95.5	93.4	85.9
	Total number eggs/plot/week	1521	1131	817
	Average No. of eggs/trap (±SE)	101.4 (±16.6)	75.4 (±22.2)	54.5 (±15.4)

## Data Availability

All data are available in the manuscript and in the Appendix A.

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
