# Peer review of "Reduction in Egg Fertility of Aedes albopictus Mosquitoes in Greece Following Releases of Imported Sterile Males"

_insects, 2021, doi:10.3390/insects12020110_

Round 1

Reviewer 1 Report

The article has been improved considerably.

In the Table 2 there are some commas that should be replaced (hatching %).

In the Figure 2, the scales of the graphs should be the same.

Thank you

Author Response

Reviewer #1

In the Table 2 there are some commas that should be replaced (hatching %).

 Table revised.

In the Figure 2, the scales of the graphs should be the same.

All figures were revised completely following all reviewers’ requests.

Reviewer 2 Report

The suggestions presented were accepted by the authors and the discussion of the article is more adjusted to the results obtained. However, some adjustments to the figures are still needed:

  1. The legends of figures and tables are incomplete.
  2. There is a missing week in Figure 3 Control A. I was not able to find out which one it would be.
  3. I suggest excluding the lines closing the figures at the top and on the right. This will improve the understanding of figure 3.
  4. I suggest increasing the height of the graphs in figure 2 and standardizing the scale, so that the difference in the number of eggs/trap becomes evident to the reader. In the way it is, in a quick reading of the graph, the reader may not notice the variation in the scale and think the number of eggs was similar.

Author Response

Reviewer #2

The legends of figures and tables are incomplete

All legends in Tables and Figures were revised.

There is a missing week in Figure 3 Control A. I was not able to find out which one it would be.

There is no missing value. This happens because in Control A the proportion of egg hatch was zero both in 46 and 47 week. So, only the first zero is displayed (in week 46).

I suggest excluding the lines closing the figures at the top and on the right. This will improve the understanding of figure 3.

All figures were revised completely following all reviewers’ requests.

I suggest increasing the height of the graphs in figure 2 and standardizing the scale, so that the difference in the number of eggs/trap becomes evident to the reader. In the way it is, in a quick reading of the graph, the reader may not notice the variation in the scale and think the number of eggs was similar.

All figures were revised completely following all reviewers’ requests.

Reviewer 3 Report

Balatsos et al. present a new version of a manuscript that has undergone important changes. The study has now been analyzed using a mixed-model approach that is appropriate for the longitudinal nature of the sampling program. A diversity of other issues have also been addressed. The key issue that the study was essentially unreplicated persists due to the experimental design and limitations in the supply of sterile male mosquitoes. These have been acknowledged by the authors in the Discussion.
I think that the manuscript requires only minor modification before it would be suitable for publication in Insects.
L28-29 should read [Delete: Our results demonstrated that] "Weekly release of 2280 to 2995 sterile male mosquitoes/hectare resulted in increased levels of egg sterility of Ae. albopictus at a single release site of 5 hectares."
L65, delete "mosquitoes" (so that singular verb agrees with subject in sentence)
L121-131, why wasn't door-to-door information provided at control site B? Did this affect sampling procedures at site B?
L160, delete "method" and comma before reference 24.
Table 2. Change commas to decimal points in Table 2.
L182, should read ....to the SIT plot...
L184-188. I suggest that you move the text on climatic conditions to the beginning of the Results section, followed by the text in lines 181-183.
L195, change to "....the median egg hatch.... (if these percentages are indeed medians)
Figure 2. This figure would benefit from three changes: (1) Increase the size of the y-axes to improve readability (2) uniformize the y-axes to the same scale for A, B and SIT (e.g. 0 - 300 in all cases), (3) Indicate the meaning of the box and whisker plots in the figure legend - to explain what bar, box, whisker and outlier points indicate.
Figure 3. Size of y-axes should be increased. The y-axis label should be changed to "Proportion of egg hatch". Legend should be changed to "Weekly proportion of egg hatching in control...." This is necessary because you use percentages not proportions in the text.
L211 should read: ."....resistance in mosquitoes, use of SIT as a tool in integrated vector control has gained increasing international interest."
L229, should read ....median egg hatch rates....
L262, should read: A major limitation....
The references are correctly formatted for Insects.
The English grammar requires a little polishing but this can be performed by the journal production staff.

Author Response

Reviewer #3

L28-29 should read [Delete: Our results demonstrated that] "Weekly release of 2280 to 2995 sterile male mosquitoes/hectare resulted in increased levels of egg sterility of Ae. albopictus at a single release site of 5 hectares."

 Text revised.

L65, delete "mosquitoes" (so that singular verb agrees with subject in sentence)

Text revised.

L121-131, why wasn't door-to-door information provided at control site B? Did this affect sampling procedures at site B?

The small size of area (Control A and SIT plots) could possibly favor the immigration of already mated females from the surrounding areas into the SIT area, a parameter which is difficult to measure. Therefore, we decided to implement the public education campaign aiming to reduce breeding sites in both plots. On the contrary, Control B, which was at a distance of 1.5 km, ensures that mosquitoes cannot be immigrated in SIT area and thus selected as an additional control.

L160, delete "method" and comma before reference 24.

Table 2. Change commas to decimal points in Table 2.

Text revised.

L182, should read ....to the SIT plot...

Text revised.

L184-188. I suggest that you move the text on climatic conditions to the beginning of the Results section, followed by the text in lines 181-183.

L195, change to "....the median egg hatch.... (if these percentages are indeed medians)

Text revised.

Figure 2. This figure would benefit from three changes: (1) Increase the size of the y-axes to improve readability (2) uniformize the y-axes to the same scale for A, B and SIT (e.g. 0 - 300 in all cases), (3) Indicate the meaning of the box and whisker plots in the figure legend - to explain what bar, box, whisker and outlier points indicate

All figures were revised completely following all reviewers’ requests.

Figure 3. Size of y-axes should be increased. The y-axis label should be changed to "Proportion of egg hatch". Legend should be changed to "Weekly proportion of egg hatching in control...." This is necessary because you use percentages not proportions in the text.

All figures were revised completely following all reviewers’ requests.

L211 should read: ."....resistance in mosquitoes, use of SIT as a tool in integrated vector control has gained increasing international interest."

Text revised.

L229, should read ....median egg hatch rates....

Text revised.

L262, should read: A major limitation....

Text revised.